# Analysis of the Twisting Tension in the Direct-Twisting Machine and the Fitting Model Based on the Experimental Data

**Shunqi Mei** [1,2]**, Mengying Zhang** [1,]*****, Di Qi** [1]**, Liye Yang** [1]**, Qiao Xu** [1,]***** and Ming Zhang** [3]

1    Hubei Digital Textile Equipment Key Laboratory, Wuhan Textile University, Wuhan 430073, China; sqmei@wtu.edu.cn (S.M.); 2015373017@mail.wtu.edu.cn (D.Q.); 1915213025@mail.wtu.edu.cn (L.Y.)
2    College of Mechanical and Electrical Engineering, Xi'an Polytechnic University, Xi'an 710048, China
3    Yichang Jingwei Textile Machinery Co., Ltd., Yichang 443001, China; ree_001@165.com
*    Correspondence: 1915033006@mail.wtu.edu.cn (M.Z.); 2006109@wtu.edu.cn (Q.X.)

**Abstract:** The tension of the balloon yarn in the direct-twisting machine affects the yarn breakage and balloon shape, which in turn affects the energy consumption and yarn productivity of the direct-twisting machine. At present, research on the balloon tension of an industrial yarn direct-twisting machine is very rare, both in China and abroad. In this regard, this paper establishes a theoretical model for balloon yarn tension during the yarn twisting process of an industrial yarn direct-twisting machine based on yarn balloon kinematics. The experiment of the yarn balloon tension of the direct-twisting machine under different yarn fineness and different spindle angular speed is carried out. The influence of the angular speed of the spindle, the yarn fineness, and other factors on the tension of the balloon yarn in the direct-twisting machine is investigated. By using mathematical statistics and regression analysis methods, the fitting equations of the yarn balloon tension of the direct-twisting machine are established. The research results show that the relationship between the balloon yarn tension of the direct-twisting machine, the angular velocity of the spindle, and the yarn fineness can be fitted by a quadratic polynomial. The predicted value of balloon yarn tension based on the fitted equation has a small relative error compared to the measured value, and the prediction result is more reliable and accurate. This paper contributes specifically by adding to the understanding of how to model the yarn tension for the specific machine and the range of the test. It also contributes to the generical methodology on how to develop such a model.

**Keywords:** direct-twisting machine; balloon yarn tension; experimental data; fitting equation

## 1. Introduction

Tire cord fabric is the skeleton material of automobile tires. As a piece of equipment used in the tire cord fabric production process, a direct-twisting machine is used to twist the cord yarn. Due to its larger packages, better quality, less waste and wire consumption, smaller footprint, higher efficiency, and smaller labor requirements, the cord direct-twisting machine has been widely used in recent years and has become one of the key pieces of equipment for cord processing [1].

When the direct-twisting machine is working, the high-speed rotation of the industrial yarn produces a balloon, which has the characteristics of high tension and high energy consumption. Under certain circumstances, the twisting tension will have an effect on the shape of the twisting balloon. The size of the balloon is related to the energy of the direct-twisting machine. The smaller the maximum radius of the balloon, the less energy it needs to consume.

For cord manufacturers, reducing twisting yarn breakage, improving twisting quality, increasing twisting production efficiency, and reducing energy consumption have always been the most important problems to be solved in the process design and optimization of direct-twisting machines. To this end, this paper investigates the variation law of balloon

yarn tension in direct-twisting machines, which can provide a basis for the analysis and design of the twisting mechanism of direct-twisting machines, and is of great significance for controlling the quality of twisting production and energy consumption.

Basic research on yarn balloon tension began in the 1950s. Mack and De Barr established a second-order differential equation system for the balloon and tension on the basis of mathematical models [2]. Zheng-Xue Tang et al. used computer data acquisition to measure yarn tension, established a numerical model of ring spinning yarn balloon movement, and discussed the influence of air resistance and balloon shape on yarn tension [3,4]. Mahmud et al. established a mathematical model of yarn tension and pellet shape varying with spindle speed during ring spinning, and used a system of nonlinear differential equations to describe the dynamic yarn path from the conveyor roller to the reel point [5]. Tang HB et al. studied the yarn dynamics and twist distribution in the improved ring spinning system, and obtained the numerical solution of the yarn path, yarn tension, and twist distribution in the steady state [6]. Li Xinrong and others established a dynamic model of the ring spinning balloon under the condition of considering Coriolis inertial force, air resistance, and yarn dynamics; deduced the boundary conditions of the balloon shape during the rotation process; and proposed an identification method for the initial parameters of the balloon shape based on a genetic algorithm [7]. Tran C D et al. proposed a numerical analysis technique based on radial basis function networks (RBFNs) for the dynamics of the toroidal spinning aerosphere. This method used the "universal approximator" based on the neural network method to solve the differential control equation derived from the yarn dynamic balance condition to determine the shape of the yarn balloon [8]. Hossain M et al. developed a frictionless superconducting magnetic-bearing twisting system, which evaluated the influence of different parameters such as spinning angular velocity, yarn count, and balloon control loop, and used variance analysis to statistically analyze the influence of these process parameters [9]. He J H et al. studied the nonlinear differential equation of the balloon without considering the air resistance, and accurately measured the maximum radius of the balloon and the maximum yarn tension [10]. Zhan Kuihua et al. studied the air ring and tension in the two-for-one twisting machine for silk and chemical fiber in silk weaving, and established differential equations for the free air ring section and the winding section of the spindle with the cylindrical coordinate system under the premise of considering the air resistance [11]. Professor Zhou Bingrong of Donghua University mainly studied the spinning balloon theory, and gave an approximate and correct solution to the balloon yarn curve without considering the air resistance and Coriolis force. The relationship between balloon shape parameters and yarn tension have been studied according to the approximate solution of sine curve [12]. Yin Rong discussed the differential equation and boundary conditions in the presence of air resistance in his paper [13]. Mao Liming and others also studied the problem of yarn tension in ring spinning and double twist spinning, and investigated the influence of balloon geometric parameters on tension [14]. Praček S studied the yarn unwinding process, established the motion equation of the unwinding yarn, and simulated the yarn unwinding [15,16].

The direct-twisting machine has been more and more widely used in the production of high-performance industrial yarn products, and the requirements for energy consumption and twisted yarn quality in the production have become higher and higher. In the field of spinning, in order to improve spinning quality and reduce energy consumption, researchers and engineers have conducted many studies on the relationship between spinning tension and balloons from the perspective of combining theory and practice from several decades, but theoretical and experimental studies on balloon yarn tension in industrial yarn direct-twisting are still very rare.

In this paper, the theoretical analysis of yarn balloon motion and tension during the twisting process of direct-twisting machine is carried out, and an experimental study of yarn balloon tension is carried out under different twisting process conditions, and then the fitting equations of the relationship between yarn tension, balloon shape, and process

parameters are established to provide reference for the analysis and design of twisting mechanism of a direct-twisting machine.

## 2. Movement and Tension Modeling of Balloon Yarn in a Direct-Twisting Machine

In the process of twisting industrial yarns, the yarns need to perform high-speed revolving movements, and their movement patterns are called "balloons". The instantaneous shape of the balloon is actually a complicated space curve. The micro-element modeling diagram of the balloon is shown in Figure 1. The differential equation of motion of balloon yarn can be established with the aid of viscoelastic mechanics and a dynamics analysis method, as shown in Figure 1, in a cylindrical system, by Equation (1) [17]:

$$
\begin{cases}
\frac{\partial}{\partial s}\left(P\frac{\partial r}{\partial s}\right) - rP\left(\frac{\partial \phi}{\partial s}\right)^2 + F_r = T\left(\frac{\partial^2 r}{\partial t^2} - \omega^2\right), \\
\frac{1}{r}\frac{\partial}{\partial s}\left(r^2 P\frac{\partial \phi}{\partial s}\right) + F_\phi = T\left[r\left(\varepsilon_x + \frac{\partial^2 \phi}{\partial t^2}\right) + 2\omega\frac{\partial r}{\partial t}\right] \\
\frac{\partial}{\partial s}\left(P\frac{\partial z}{\partial s}\right) + F_z = T\frac{\partial^2 z}{\partial t^2} \\
\left(\frac{\partial r}{\partial s}\right)^2 + \left(r\frac{\partial \phi}{\partial s}\right)^2 + \left(\frac{\partial z}{\partial s}\right)^2 = 1 \\
\omega = \omega_x + \frac{\partial \phi}{\partial t}, \varepsilon_x = \frac{\partial \omega_x}{\partial t}, \omega_x = \omega_x(t)
\end{cases}
\tag{1}
$$

where $t$ is the time, s; $s$ is the arc coordinate (Euler coordinate) of a certain point on the balloon yarn, m; $r$, $\varphi$, and $z$ are the cylindrical coordinates of the point on the balloon yarn, m, rad, and m, respectively; $P$ is the yarn tension at any point of balloon, N; $T$ is the yarn fineness, tex ($10^{-6}$ kg/m).

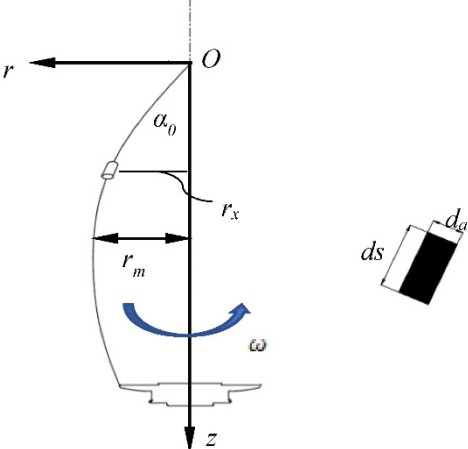

**Figure 1.** Micro-element modeling diagram of the balloon.

$F_r$, $F_\varphi$, and $F_z$ are the projections of the sum of external forces acting on the balloon yarn in the axes of the cylindrical coordinate system, N; $\omega_x$ is the angular velocity of the moving rectangular Cartesian coordinate system, in which the origin $O$ and the axis $z$ coincide with the origin $O$ and the axis $z$ of the moving cylindrical coordinate system $Or\varphi z$, rad/s; and $\omega$ is the angular velocity of the balloon yarn, rad/s. $d_a$ is the theoretical diameter of the yarn, m and the actual size of the yarn diameter is generally not considered in the analysis.

In order to meet the needs of engineering analysis and calculation, Equation (1) can be simplified and solved, and the general expression of balloon yarn tension can be obtained, as follows:

$$
P_0 = f_p(T, r_m^2, \omega^2, \alpha_0)
\tag{2}
$$

$P_0$ is the yarn tension at the point $O$, N; $r_m$ is the maximum radius of balloon, m; $\alpha_0$ is the top angle of balloon, rad; $f_p$ represents the function between $P_0$ and $T$, $r_m$, $\omega$, and $\alpha_0$.

The above formula shows the basic change law between balloon tension and balloon geometric parameters and motion parameters. As one of the typical application cases, in the twisting process of ring spinning, the balloon has the characteristics of elongated shape and small balloon tip angle. The balloon contour curve is similar to a sine curve. After simplified processing, the yarn tension can be obtained as shown in Equation (3). This equation also has a certain reference for studying the yarn balloon of a direct-twisting machine [18].

$$P_0 = \frac{1}{4} Tr_m^2 \omega^2 \left[ 1 + 1/\tan^2(\alpha_0/2) \right] \tag{3}$$

## 3. Experiment and Result Analysis

In order to investigate the change law of balloon yarn tension during the twisting process of the direct-twisting machine, the tension of the balloon yarn was tested under different technological conditions.

The test model is shown in Figure 2. The test equipment included a K3501 direct-twisting machine from Yichang Jingwei Textile Machinery Co., Ltd., Yichang, China. A PHANTOM V711 high-speed camera and DSS-10LED high-precision flash speedometer were used to measure the instantaneous shape and instantaneous speed of the balloon. In order to optimize the imaging effect of high-speed cameras, high-brightness LED spotlights were used to fill light. The DTMB-5000SCHMID model tension meter was used to measure the instantaneous tension of the balloon yarn.

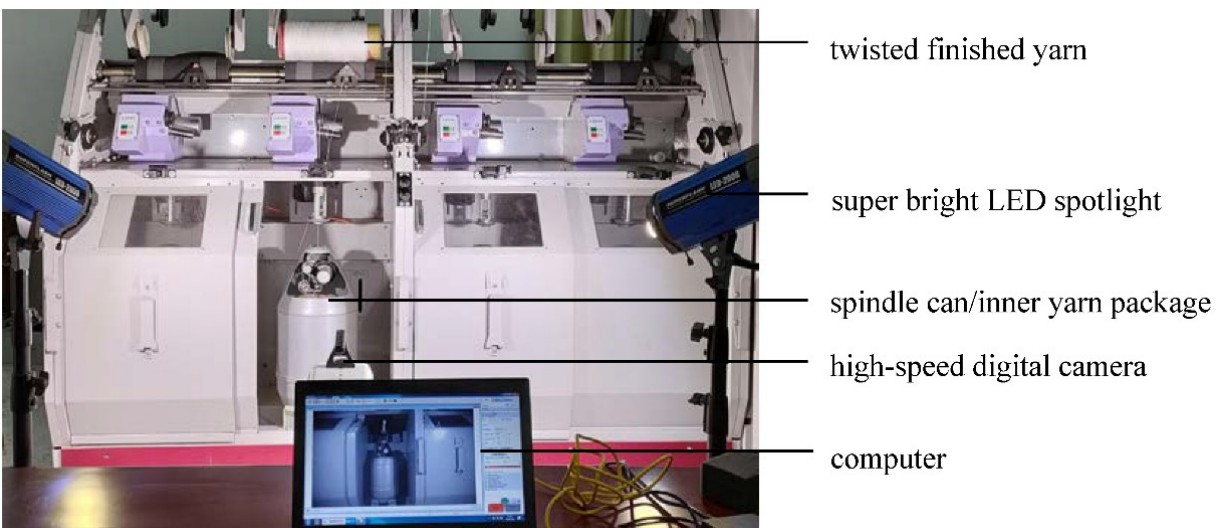

twisted finished yarn

super bright LED spotlight

spindle can/inner yarn package

high-speed digital camera

computer

**Figure 2.** Experimental equipment for the direct-twisting machine.

The industrial yarn for the test included: 930 dtex of nylon, 1100 dtex, 1440 dtex, 1670 dtex, 2200 dtex of polyester, and 1 dtex = 0.1 tex = $10^{-7}$ kg/m. The relative position of the above device is shown in Figure 2.

The structure diagram of the direct-twisting mechanism is shown in Figure 3, where $h_1$ = 1 mm, $h_2$ = 15 mm, and $h_3$ = 38 mm. The diameter of the spindle can is D, which is 266 mm, and the height of the balloon H is 595 mm. Figure 4 shows the balloon morphology of the direct-twisting machine taken by the high-speed digital camera during the experiment under high-speed operation.

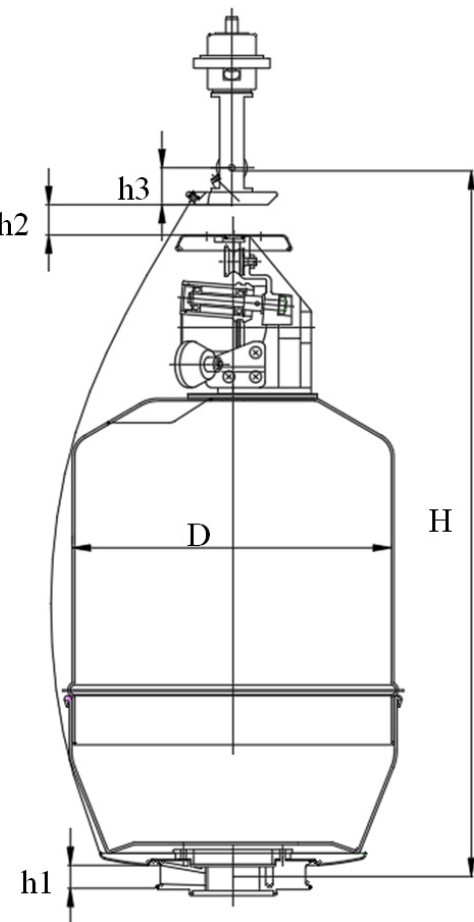

**Figure 3.** Schematic diagram of direct-twisting mechanism.

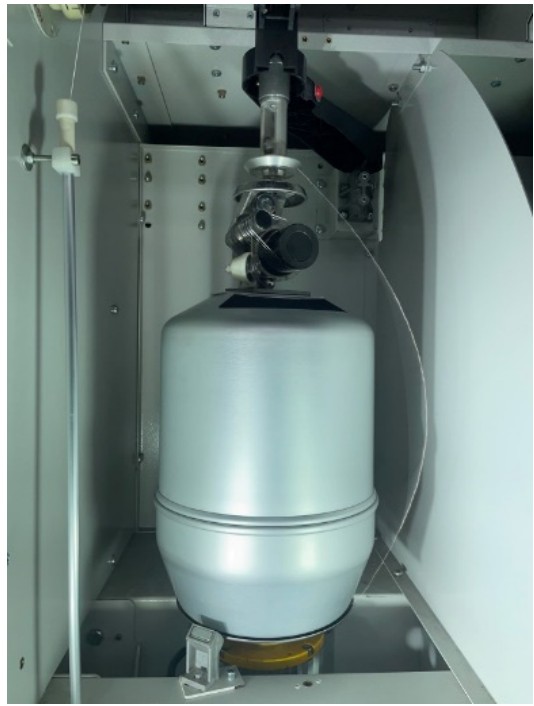

**Figure 4.** Transient shape of the balloon.

### 3.1. Influence of Spindle Angular Speed on Twisting Tension

In order to investigate the influence of the rotation speed of the balloon on the tension of the balloon yarn, the twisting tension of the balloon yarn was tested under different spindle angular velocities. For the convenience of measurement, the tension between the twister and the overfeeding roller is measured as the twisting tension of the balloon yarn. The measurement principle of twisting tension is shown in Figure 5.

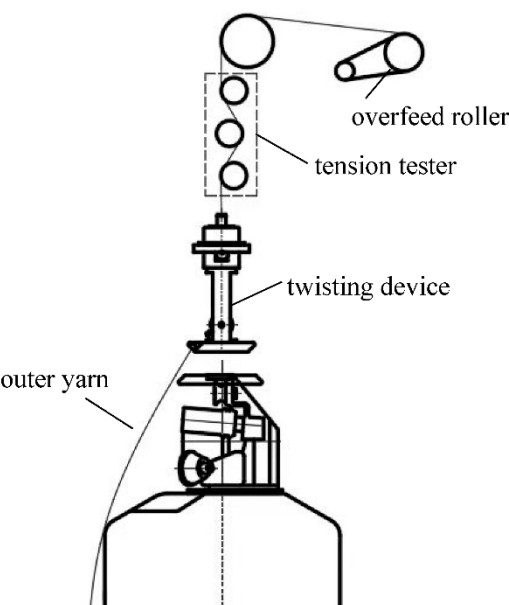

**Figure 5.** Principle of balloon yarn tension measurement.

The configuration of the twisting assembly during the test were: the diameter of the twisting tray was 220 mm, the diameter of the yarn storage tray was 120 mm, and the diameter of spindle can was 266 mm. When using the tension meter DTMB-5000SCHMID for the tension test of the direct-twisting machine, in order to reduce the random measurement error as much as possible, five tension measurement values were read under a certain measurement condition, and the average value was taken as the tension measurement value under the measurement condition. For the fineness of the twisted yarn was 1440 dtex, the measurement results of twisting yarn tension at different spindle speeds are shown in Table 1. The standard deviation and the overall standard deviation coefficient of measurement data in each group are calculated. The results show that the standard deviation of tension measurement values in each group fluctuates in the range of 3 cN–5 cN, 1 cN = 0.01 N, and the overall standard deviation coefficient is in the range of 0.22–0.27%, indicating that the method of measuring the twisting tension in this paper is feasible and the measurement results are accurate. Meanwhile, in the subsequent tension measurement tests in this paper, the above method was used to obtain the tension measurement values.

The experiment's data show that the twisting tension had obvious regularity, and it increased with the increase of the spindle angular velocity. To establish the relationship between twisting tension $P_\omega$ and spindle angular velocity $\omega$, on the basis of experimental measurement data, a polynomial fitting method was used for fitting, and the relationship equation was obtained as shown in Equation (4). The tension–spindle angular velocity data distribution and fitting curve measured by the experiment are shown in Figure 6. The unit of $P_\omega$ is cN, the unit of $\omega$ is rad/s.

$$P_\omega = 0.00467\omega^2 - 5.72722\omega + 3032.47 \tag{4}$$

**Table 1.** Tension measurement results and standard deviation analysis of different spindle angular velocity.

| Experimental Item | Spindle Angular Velocity, r/Min (rad/s) | The Measurement of Twisting Tension (cN) | The Mean of Twisting Tension (cN) | Standard Deviation (cN) | Coefficient of Overall Standard Deviation |
|---|---|---|---|---|---|
| 1 | 7000 (733.04) | 1341.2 1346.7 1339.8 1347 1349.3 | 1344.8 | 3.65 | 0.27% |
| 2 | 7500 (785.40) | 1421 1413.9 1423 1416 1414.1 | 1417.6 | 3.72 | 0.26% |
| 3 | 8000 (837.76) | 1519 1523.1 1515.4 1524.8 1520.7 | 1520.6 | 3.27 | 0.22% |
| 4 | 8500 (890.12) | 1634 1629.2 1627.8 1635.9 1624.1 | 1630.2 | 4.26 | 0.26% |
| 5 | 9000 (942.48) | 1794.4 1785.3 1783.5 1796.1 1787.7 | 1789.4 | 4.99 | 0.28% |

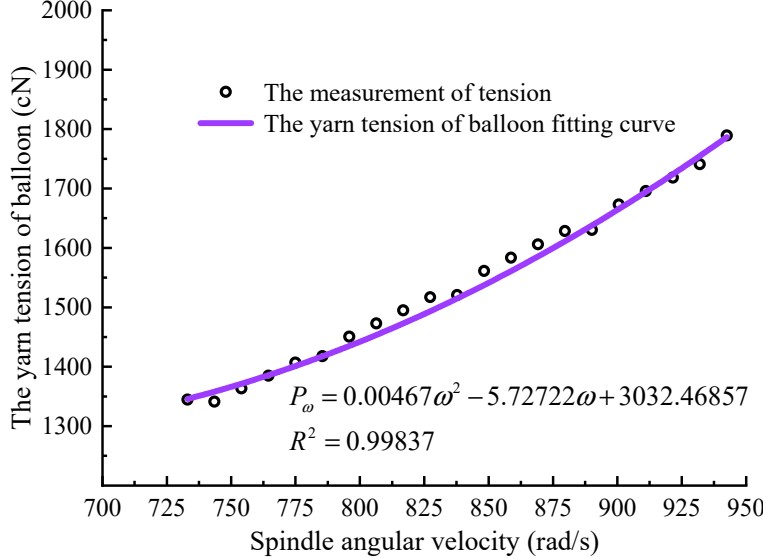

**Figure 6.** Tension–spindle angular velocity fitting curve for 1440 dtex.

### 3.2. Variation Law of Twisting Yarn Tension under Different Yarn Fineness

In order to investigate the effect of yarn fineness on the twisting tension, in the experiment, the twisting tension was measured for yarns of different fineness. The polynomial fitting method was used for fitting, and the fitting equations of the relationship between

twisting tension and spindle angular velocity under different fineness were obtained, as shown in Table 2, and the fitting curve is shown in Figure 7. $P_{\omega 1}$, $P_{\omega 2}$, $P_{\omega 3}$, $P_{\omega 4}$, and $P_{\omega 5}$ represent the fitting values of twisting tension obtained under different fineness, the unit of $P_{\omega 1}$–$P_{\omega 5}$ is cN. $R^2$ represents the coefficient of determination.

**Table 2.** Statistics table of fitting equation for twisting tension and spindle angular velocity under different yarn fineness.

| Yarn Fineness | Fitting Equation | $R^2$ |
|---|---|---|
| 930 dtex | $P_{\omega 1} = 0.00438\omega^2 - 6.44385\omega + 3493.50$ | 0.99152 |
| 1100 dtex | $P_{\omega 2} = 0.00326\omega^2 - 3.91692\omega + 2422.11$ | 0.99676 |
| 1440 dtex | $P_{\omega 3} = 0.00467\omega^2 - 5.72722\omega + 3032.47$ | 0.99837 |
| 1670 dtex | $P_{\omega 4} = 0.00131\omega^2 - 0.20779\omega + 1393.84$ | 0.99963 |
| 2200 dtex | $P_{\omega 5} = 0.00283\omega^2 - 0.75535\omega + 1075.35$ | 0.99410 |

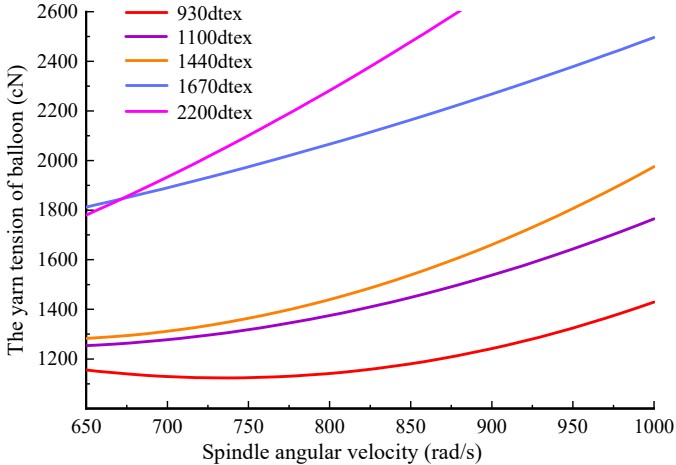

**Figure 7.** Fitting curves of balloon yarn tension and spindle angular velocity under different yarn fineness.

The experimental conditions: industrial yarn fineness were: 930 dtex, 1100 dtex, 1440 dtex, 1670 dtex, 2200 dtex, the diameter of twisting tray was 220 mm, the diameter of yarn storage tray was 120 mm, and the diameter of spindle can was 266 mm.

It can be analyzed from the measurement results and the fitting curve that when the angular velocity of the spindle increased from 680.68 rad/s (6500 r/min) to 994.84 rad/s (9500 r/min), the twisting tension gradually increased. At the same angular velocity, the twisting tension increased in turn with the increase in yarn fineness. Yarn with a fineness of 2200 dtex had the highest twisting tension, and the yarn with a fineness of 930 dtex has the lowest twisting tension.

### 3.3. Binary Fitting of Twisting Tension to Spindle Angular Speed and Yarn Fineness

The spindle angular speed of the direct-twisting machine and the fineness of the industrial yarn were the key factors that affect the twisting tension. In order to comprehensively investigate the influence of these two factors on the tension, different twisting components were selected, and the spindle angular velocity and yarn fineness were changed, so as to obtain the experimental data of twisting tension under the conditions of different spindle angular speed and yarn fineness.

For example, when the configuration of the twisting tray/yarn storage tray/spindle can was: 220, 120, and 266 mm, and the fineness of the industrial yarn were: 930, 1100, 1440, 1670, and 2200 dtex, the twisting tension was measured under a series of spindle angular velocity, as shown in Table 3.

**Table 3.** Measurement values of twisting tension under different yarn fineness and spindle angular velocity.

| Twisting Configuration | Experimental Item | Spindle Angular Velocity (rad/s) | Yarn Fineness (dtex) | Twisting Tension (cN) |
|---|---|---|---|---|
| | 1 | 837.76 | 930 | 1193.0 |
| | 2 | 890.12 | 930 | 1208.0 |
| | 3 | 942.48 | 930 | 1355.0 |
| | 4 | 785.40 | 1100 | 1361.0 |
| | 5 | 837.76 | 1100 | 1427.4 |
| | 6 | 890.12 | 1100 | 1526.6 |
| | 7 | 942.48 | 1100 | 1628.8 |
| | 8 | 733.04 | 1440 | 1344.8 |
| twisting tray/yarn storage | 9 | 785.40 | 1440 | 1417.6 |
| tray/spindle can: | 10 | 837.76 | 1440 | 1520.6 |
| 220 mm/120 mm/266 mm | 11 | 890.12 | 1440 | 1630.2 |
| | 12 | 942.48 | 1440 | 1789.4 |
| | 13 | 733.04 | 1670 | 1948.6 |
| | 14 | 785.40 | 1670 | 2036.2 |
| | 15 | 837.76 | 1670 | 2145.8 |
| | 16 | 890.12 | 1670 | 2247.8 |
| | 17 | 733.04 | 2200 | 2007.4 |
| | 18 | 785.40 | 2200 | 2268.8 |
| | 19 | 837.76 | 2200 | 2415.6 |

According to the data in Table 3, the twisting tension increased with the increase of spindle angular speed and yarn fineness, and the variation of twisting tension had obvious regularity. In order to fit the binary equation of the twisting tension $P_d$, the spindle angular velocity $\omega$ and the yarn fineness $T$ of the direct-twisting machine, the Levenberg–Marquardt algorithm of the origin software was used to non-linearly fit the curved surface. The spindle angular velocity $\omega$ and yarn fineness $T$ were selected as the independent variables. Twisting tension $P_d$ was selected as the dependent variable. According to the measurement data in Table 3 above, various nonlinear fitting equations were obtained, which are numbered 1–8, as shown in Table 4. $P_{d1}$, $P_{d2}$, $P_{d3}$, $P_{d4}$, $P_{d5}$, $P_{d6}$, $P_{d7}$, and $P_{d8}$ represent the fitting values of twisting tension under different binary fitting equations. The unit of $P_{d1}$–$P_{d8}$ is cN. $m_1$ and $m_2$ are the intermediate variables of the transition.

**Table 4.** Binary fitting equations of balloon yarn tension of the direct-twisting machine.

| Number | Fitting Equation | $R^2$ |
|---|---|---|
| 1 | $P_{d1} = 1.52 \times 10^7 \cdot e^{m_1} - 1.47 \times 10^{-4}T^2 + 1.26T + 211.48,\ m_1 = 1 - e^{\frac{9.88 \times 10^8 - \omega}{8.3 \times 10^7}}$ | 0.66834 |
| 2 | $P_{d2} = -0.00446\omega^2 + 8.79241\omega - 1.82192 \times 10^{-4}T^2 + 1.14944T + 4.60594 \times 10^{-4}\omega T - 4315.98$ | 0.80733 |
| 3 | $P_{d3} = -8158.42947 e^{m_2} + 4721.93,\ m_2 = \frac{\omega}{1588.912} - \frac{T}{3060.676}$ | 0.81977 |
| 4 | $P_{d4} = 1.91396\omega + 0.95618T - 1286.85$ | 0.82736 |
| 5 | $P_{d5} = 0.00656\omega^{0.96419} \times T^{0.82311}$ | 0.83979 |
| 6 | $P_{d6} = \dfrac{-1.038 \times 10^6\omega + 167228.12T - 86.9485T^2 + 235.31\omega T + 4.125 \times 10^8}{-451.902\omega - 0.10452\omega^2 + 254.79T - 0.10118T^2 + 0.18961\omega T + 1}$ | 0.86190 |
| 7 | $P_{d8} = -8.37 \times 10^{-10}\omega^5 + 4.19 \times 10^{-6}\omega^4 - 0.00813\omega^3 + 7.656\omega^2 - 3525.9\omega$ $-7.2 \times 10^{-12}T^5 + 4.069 \times 10^{-8}T^4 - 8 \times 10^{-5}T^3 + 0.05989T^2 - 4.705T + 629748.35$ | 0.98081 |
| 8 | $P_{d8} = -0.00407\omega^2 - 7.15227 \times 10^{-4}T^2 + 8.92584\omega + 3.19332T - 5816.53$ | 0.98601 |

The accuracy of the fitting equations was measured by the coefficient of determination $R^2$, which is also called the goodness of fit. For a set of sample spaces containing $N$-ordered pairs of real numbers, $(x_i, y_i)$ ($i = 1, 2, 3, \ldots, N$) is the sample points, $\overline{y}$ is the average of the

sample ordinates, and $y'$ is the fitting of $x_i$ value, the coefficient of determination is shown in Equation (5) [19,20].

$$R^2 = 1 - \frac{\sum\limits_{i=1}^{N}(y_i - y')^2}{\sum\limits_{i=1}^{N}(y_i - \overline{y})^2} \quad (5)$$

The coefficient of determination $R^2$ value range is [0, 1]. The closer $R^2$ is to 1, the better the fit of the regression line to the observed values; Conversely, the closer $R^2$ is to 0, the worse the fit of the regression line to the observed values [21].

It can be seen from the Table 4 that the coefficient of determination $R^2$ of the fitting equation of No. 1 was the smallest, which was 0.66834, indicating that the empirical equation of No. 1 had a poor fitting degree. The fitting equation of No. 8 had the largest value of determination coefficient $R^2$, which was 0.98601, indicating that the fitting degree was relatively good. By comparing the fitting equations, the fitting equation with the largest coefficient of determination $R^2$ was selected as the best fitting model.

Therefore, the fitting equation of the relationship between balloon yarn tension, spindle angular velocity, and yarn fineness is shown in Equation (6), where the coefficient of determination $R^2 = 0.98601$. The fitting surface obtained according to the fitting equation is shown in Figure 8. The unit of $P_d$ is cN.

$$P_d = -0.00407\omega^2 - 7.15227 \times 10^{-4}T^2 + 8.92584\omega + 3.19332T - 5816.53 \quad (6)$$

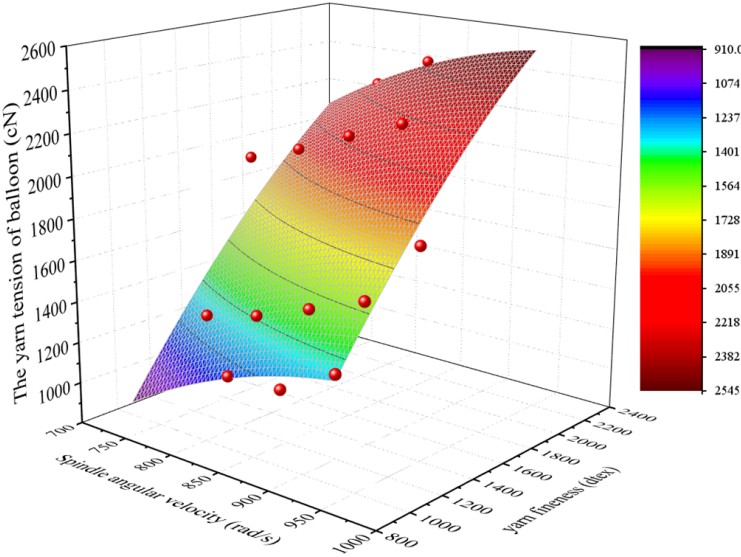

**Figure 8.** Fitting surface of the relationship between yarn tension of the balloon, spindle angular velocity, and yarn fineness.

### 3.4. Comparative Analysis of Fitting Equation and Measured Value

According to the fitting equation obtained above, the twisting tension of the direct-twisting machine under different process conditions can be predicted, and the results of the comparison with the measured values are shown in Table 5.

**Table 5.** Measured and predicted twisting tension error analysis.

| Twisting Configuration | | | Yarn Fineness/dtex | Spindle Angular Velocity/(rad/s) | Measured Value/cN | Predicted Value/cN | Error Analysis | |
|---|---|---|---|---|---|---|---|---|
| Twisting Tray/mm | Yarn Storage Tray/mm | Spindle Can/mm | | | | | Absolute Error/cN | Relative Error/% |
| | | | 930 | 942.48 | 1355.0 | 1329.0 | 26.0 | 1.92 |
| | | | 1100 | 785.40 | 1361.0 | 1339.9 | 21.1 | 1.55 |
| | 220 | | 1100 | 837.76 | 1427.4 | 1449.1 | −21.7 | −1.52 |
| | | | 1100 | 890.12 | 1526.6 | 1547.8 | −21.2 | −1.39 |
| | 120 | | 1100 | 942.48 | 1628.8 | 1624.3 | 4.5 | 0.28 |
| | | | 1670 | 785.40 | 2036.2 | 2018.9 | 17.3 | 0.85 |
| | 266 | | 1670 | 837.76 | 2145.8 | 2140.0 | 5.8 | 0.27 |
| | | | 1670 | 890.12 | 2247.8 | 2238.8 | 9.0 | 0.40 |
| | | | 2200 | 785.40 | 2268.8 | 2244.3 | 24.5 | 1.08 |

It can be seen from the above table that the predicted value obtained by using the fitting equation in this paper was very close to the actual measured value. For example, when the configuration of the twisting tray/yarn storage tray/spindle was: 220, 120, and 266 mm, and the yarn fineness was 930, 1100, 1440, 1670, 2200 dtex, the absolute value of the absolute error about the twisting tension was 4.5 cN~26.0 cN, and the average value was 7.2 cN, the absolute value of the relative error was 0.28~1.92%, and the average was 0.38%. The results show that the research method and the established fitting equation of the twisting tension of the direct-twisting machine in this paper are feasible.

## 4. Conclusions

In the twisting process of the direct-twisting machine, the twisting tension is the key factor affecting the balloon shape, twisting quality, and twisting energy consumption. In the case of ignoring air resistance, the tension equation of the yarn balloon can be established according to the principle of kinematics. Through this theoretical equation, the relationship between tension and process parameters such as spindle angular velocity and yarn fineness can be roughly seen. However, the exact relationship between tension and process parameters such as yarn fineness and spindle angular speed for a specific model of yarn balloon needs to be predicted by fitting equations based on experimental data. It is an effective method to establish the fitting equation of tension change based on the test data, which can provide a more accurate reference basis for the design of twisting mechanism and process parameters of a direct-twisting machine.

**Author Contributions:** Conceptualization, S.M. and Q.X.; Methodology, S.M., M.Z. (Mengying Zhang), L.Y. and Q.X.; Software, M.Z. (Mengying Zhang) and D.Q.; Validation, S.M. and M.Z. (Mengying Zhang); Formal analysis, M.Z. (Mengying Zhang) and L.Y.; Investigation, M.Z. (Mengying Zhang), D.Q., M.Z. (Ming Zhang) and L.Y.; Resources, S.M. and M.Z. (Ming Zhang); Data curation, M.Z. (Mengying Zhang), D.Q. and L.Y.; Writing—original draft preparation, M.Z. (Mengying Zhang) and S.M.; Writing—review and editing, S.M. and M.Z. (Mengying Zhang); Visualization, S.M. and M.Z. (Mengying Zhang); Supervision, S.M.; Project administration, S.M., Q.X. and M.Z. (Ming Zhang); Funding acquisition, S.M. All authors have read and agreed to the published version of the manuscript.

**Funding:** This work was supported by The Science and Technology Program of Hubei Province (No. 2019AEE011) and The National Science Foundation of China of China (No. 51175385).

**Institutional Review Board Statement:** Not applicable.

**Informed Consent Statement:** Not applicable.

**Data Availability Statement:** Data is contained within the article.

**Conflicts of Interest:** The authors declare no conflict of interest.

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
