# Peer review of "Analysis of the Twisting Tension in the Direct-Twisting Machine and the Fitting Model Based on the Experimental Data"

_applsci, doi:10.3390/app12094298_

Round 1

Reviewer 1 Report

This work concerns the theoretical and experimental analysis of the direct-twisting machine and the corresponding fitting model based on the experimental data. Some elements of the research reported in this paper are unclear and need improvement. 

Particularly, the Authors solve the equations system (1) but there is no precise information, on how it has been achieved - is this numerical solution or some general/particular analytical solutions are available. 

Experimental testing needs some experimental error discussion together with experimental statistics (mean value, standard deviation, etc.) to see how precise these experiments really are. 

The fitting procedure in the case of a single parameter function seems to be the rather efficient, however two-parametric case is rather distant from the set of solutions/measurements. So, the table showing fitting error needs to be attached together with various polynomials. It was not described in detail what type of the Least Squares Method has been used and why the Authors prefer specifically polynomial bases. Maybe some other functions could be more efficient in this particular case study, cf. On selecting composite functions based on polynomials for responses describing extreme magnitudes of structures by B. PokusiÅ„ski, M. KamiÅ„ski, Appl. Sci. 202111(21), 10179; https://doi.org/10.3390/app112110179 - 30 Oct 2021. 

It is hard to agree with the conclusion that (cit.): "The yarn balloon motion equation and tension change equation can be established by using the principles of dynamics and viscoelasticity" because this does not follow research findings and theoretical apparatus presented in this work. 

Reviewer 2 Report

Review applsci-1635339

Analysis of the Twisting Tension in the Direct-twisting Ma-chine and the Fitting Model based on the Experimental Data

The authors address an interesting research topic for the journal Applied Sciences. Anyway, some recommendations should be taken into account:

  • The abstract should be expanded to include the objectives of the research work as well as some reference to the most interesting contributions that can be found in the article.
  • Given that the magnitudes worked in the article exceed one thousand units, why not use N instead of cN? This aspect affects several parts of the paper (Table 1, equations of Table 2, Figures 6-8,….). However, from my point of view, this change improved the paper.
  • Please use the same number of decimal places for similar concepts (e.g. data in Table 2, Table 4,... need to be modified).
  • Please, clarify the units for equation 5.
  • Please, include in Figure 6 (figure caption) that the yarn fineness is 1440 dtex.
  • The number of references is not adequate. Please, improve the state-of-the-art in the main text.
  • Furthermore, it would be advisable to include some papers from the journals of MDPI editorial related to the topic of the manuscript (Applied Sciences, Metals, Materials, etc.).

Round 2

Reviewer 1 Report

The Authors have made the additional modifications in their text and now it is ready for publication in this journal. 

Author Response

Thank you very much for your help for our paper! We hope that the revised version of the manuscript is now acceptable for publication in this journal.

Reviewer 2 Report

Although the authors have responded to the previous review by saying that they had specified the units for all the variables in the different fitting equations, I am unable to find where they have done so.

Author Response

We are very sorry for our careless writing. We have now specified units for all variables in the different fitting equations and marked them with yellow fill.

Please see our latest revised version of the manuscript.